# Exploring Impacts of a Nutrition-Focused Massive Open Online Course

**DOI:** 10.3390/nu14183680

**Published:** 2022-09-06

**Authors:** Melissa Adamski, Helen Truby, Christie Bennett, Simone Gibson

**Affiliations:** 1Department of Nutrition, Dietetics and Food, School of Clinical Sciences, Faculty of Medicine, Nursing and Health Sciences, Monash University, Melbourne 3186, Australia; 2School of Human Movement and Nutrition Sciences, Faculty of Health and Behavioural Sciences, University of Queensland, Brisbane 4067, Australia

**Keywords:** online education, healthy eating, health promotion, nutrition education, massive open online course

## Abstract

The nutrition education landscape is changing due to advances in technology. Massive Open Online Courses (MOOCs) are an example of new education opportunities, made possible by advances in online learning environments. This research aimed to evaluate a nutrition-focused MOOC, applying Kirkpatrick’s theoretical model of learning, to comprehensively describe learners’ reactions, knowledge and behaviours. A mixed-methods approach explored learners’ experiences of participating in a global nutrition MOOC. Quantitative survey data, using descriptive statistics, measured pre-course nutrition knowledge, post-course satisfaction and learning, and changes to fruit/vegetable intake. Qualitative data from discussion forums and semi-structured interviews were thematically analysed and explored participant experiences and perceived impacts on dietary behaviours and nutrition knowledge. All results were mapped to Kirkpatrick’s model. Surveys measuring perceived knowledge, satisfaction, and fruit and vegetable intake were completed by 4941, 1003 and 1090 participants, respectively. Overall reactions to the course were positive. Perceived vegetable intake increased for 62% of participants. Twelve interviews and six hundred and forty-seven forum comments were analysed, identifying eight themes highlighting changes to knowledge and dietary behaviours, and the importance of peer-to-peer learning. All results mapped to Levels 1–3 of Kirkpatrick’s model. MOOCs can be an effective platform to communicate evidence-based nutrition information to large, global audiences.

## 1. Introduction

The nutrition education landscape is changing. Once the public accessed nutrition information and advice from magazines, books and visits to health professionals. Now, the internet and social media provide instant access to a plethora of nutrition information, often lacking evidence about its quality or scientific validity [1]. This shift in nutrition information seeking behaviour requires nutrition professionals to respond by adapting and innovating how they disseminate information [1,2]. 

Massive Open Online Courses (MOOCs) are an example of innovate delivery of education, made possible by advances in technology and online learning environments [3]. They provide opportunities for the general public to access and learn from academic experts in flexible, interactive, online learning platforms, with no to little pre-requisite learning [4,5]. MOOCs not only provide access to leading academic education institutions, but learners become part of an online learning community. Peer-to-peer sharing of information and learning from each other are major features of MOOCs, which are purposely designed to promote engagement of large numbers of learners all participating in learning at a distance [6]. Participants are encouraged to develop supportive learning communities, discussing information, asking and answering each other’s questions, which may be moderated by online mentors [3,7]. This engagement differs from more traditional didactic modes of education where information is disseminated using a directive approach (from experts to participants), where experts are in charge of what is taught and how it is learnt [3]. 

This transformation of educational structures may be uncomfortable for experts to accept, as ‘learners’ begin to assume the role of ‘teacher’, and learners take more accountability for their learning [1,8]. This shift may challenge educators to reflect on their traditional pedagogical models and consider the role of learner-led and peer-to-peer learning. MOOCs draw their inspiration from connectivist learning theory, which encourages learners to have conversations with each other and incorporates the influences of technology on learning behaviours [9]. Connectivist learning theory recognises learning occurs through interactions and connections with learning communities (typically online), and people construct their own meaning from a range of information sources [9,10]. With evolving technology and corresponding changes in people’s nutrition information seeking behaviours, nutrition professionals need to consider how they can optimise technology to reach very large audiences and compete with those online influencers who disseminate misinformation [2]. Online influencers can provide a range of views on nutrition, sometimes conflicting with evidence-based guidelines and spreading dietary misinformation and may come from backgrounds without scientific nutrition training [11,12,13,14,15]. 

MOOCs offer nutrition professionals an opportunity to promote themselves and their science to new audiences and facilitate online learning communities [16]. A range of food and nutrition-focussed MOOCs are available, providing evidence-based food and nutrition information to the public and helping combat dietary misinformation [17,18,19]. 

To date, MOOCs as a pedological method of delivering nutrition advice are not well evaluated; course completion rates vary and are typically low, with substantial heterogeneity of learners’ backgrounds and motivations for participation [20,21,22,23]. Evaluation strategies beyond pre/post course surveys have been recommended for more robust comprehensive evaluations [21,22,23]. These include mixed methods using automated sentiment analysis of MOOCs’ discussion forums and compared participant feedback response comments [24]. The MOOC evaluation protocols by Meinert et al. (2018) and Smith Lickess et al. (2019) outline mixed methods approaches, including collecting participant interview data, utilising the Kirkpatrick and RE-AIM frameworks for learning and impact evaluation [21,22]. The Kirkpatrick Model is an established model of training evaluation and has four levels of assessment: Reaction (Level 1), Learning (Level 2), Behaviour (Level 3) and Results (Level 4) [25]. 

A number of nutrition MOOCs have previously been reviewed and evaluated; however, they have relied on pre/post course survey data [17,18,19]. In 2020, we explored our own nutrition and food MOOC, a free 3-week online course called Food as Medicine, which teaches learners the role of food in health, and helps them explore how they can make dietary changes [17]. Here, the MOOC was evaluated using pre- and post- course surveys, with recommendations identified for more robust evaluation processes to determine improvements in nutrition literacy and change dietary behaviours [17].

To bridge this evaluation gap, this research applied the Kirkpatrick Model of learning, outlined in published MOOC evaluation protocols, to comprehensively evaluate the Food as Medicine MOOC. The research aimed to evaluate Levels 1–3 Kirkpatrick’s theoretical model of learning to describe learners’ reactions, attitudes, knowledge and behaviours, after participating in the Food as Medicine MOOC [22,25,26].

## 2. Materials and Methods

A mixed-methods approach was utilised to explore learners’ experiences of taking part in the MOOC, incorporating real world data (RWD) from discussion forums, as well as semi-structured interviews and purpose-built surveys [27]. This research blended various methods, across multiple runs of the MOOC to provide a more comprehensive picture [27,28]. Methods were developed with a pragmatic approach; methods built upon each other as analysis was conducted and implemented in subsequent courses. 

The inclusion of individual qualitative interviews in addition to quantitative surveys to explore the levels of the Kirkpatrick Model and impacts on learning have been outlined in MOOC evaluation protocols [21,22]. This research also included user-generated data extracted from the discussion forums, which enabled us to observe participants naturally interacting and sharing information with other learners. These multiple sources of qualitative, quantitative and user-generated data enabled triangulation of findings via the multiple source procedure [22,29]. Data were collected systematically to ensure that all 3 levels of Kirkpatrick’s model were available (see Figure 1 and Table 1) and to enable triangulation of findings from multiple sources.

### 2.1. MOOC Evaluation Protocol Applying Kirkpatrick’s Framework 

A number of published papers that evaluated MOOCs have utilised the Kirkpatrick model for evaluating learning efficacy and adoption [21,22]. The Kirkpatrick Model is an established model of training evaluation and has four levels of assessment: Level 1: Reaction (Satisfaction, Relevance, Engagement), Level 2: Learning (Commitment, Confidence, Knowledge), Level 3: Behaviour (application or utilisation of what was learnt) and Level 4: Results (degree to which targeted outcomes occur as a result) [25]. This model has been revised to the New World Kirkpatrick Model (NWKM) which has expanded scope and flexibility [30] and has been used in the evaluation of MOOCs that aim to provide training to professionals [26] and to evaluate nutrition and health education on social media [31]. This research mapped the MOOC evaluation methods to the NWKM levels (Table 1). 

### 2.2. Data Collection

A mixed methods data collection plan was developed based on Kirkpatrick’s model, data collection needed to take into consideration the nature of open access platforms and requirements by the host and the Ethics Committee who provided approval for this novel research. Data were collected from a 3-week, evidence-based nutrition MOOC developed by nutrition and dietetic academics from Monash University, called Food as Medicine. The MOOC was targeted to the general public with no pre-requisites of knowledge or prior learning required [17]. Food as Medicine has run 18 times since its first run in May 2016 and has enrolled over 230 thousand people [32]. The aim of the MOOC was to increase learner’s knowledge around foods and dietary patterns conducive to improving health, and to nudge participant’s dietary behaviours to align more with government dietary guidelines. Food as Medicine covered a range of topics on food, nutrition and health, focusing on the components of food which exert health benefits, different foods and diets involved in prevention and management of certain conditions, and how people can implement new dietary behaviours (Appendix A). 

### 2.3. Quantitative Data 

All learners were invited to complete quantitative surveys, with consent obtained to use results as part of research. All surveys were developed by content experts and piloted for face validity and functionality before implementation. The surveys were designed to be very short in duration so as not to detract from the learning experience. It was not possible to match pre- and post- survey responses due to the mandated anonymous nature of any MOOC. 

#### 2.3.1. Pre-Course Surveys

All learners were invited to complete an optional pre-course survey in three consecutive runs of the MOOC in March, June and November 2020. The pre-course survey explored learner’s nutrition education backgrounds and was offered to learners in the first step of the course (Appendix A).

#### 2.3.2. Post-Course Surveys

All learners were also invited to participate in the post-course survey which was located in the last step of the MOOC. This survey was in three consecutive runs of the MOOC in March, June and November 2020 and collected data about learners’ experiences of the MOOC, particularly around satisfaction levels and perceptions of behaviour change (Appendix A). 

#### 2.3.3. Fruit and Vegetable Survey

To gain insight as to whether the MOOC had achieved its overarching purpose of changing the dietary behaviours of learners, data was collected on participants self-reported intake of fruit and vegetables after completing the MOOC. The survey was developed by academic dietitian researchers and asked questions regarding self-perceived changes in serves, and total serves of fruit and vegetable intake per day (Appendix A).

### 2.4. Qualitative Data 

The open nature of the online MOOC discussions allowed for observation of how learners interacted, discussed and shared the course information. Semi-structured interviews enabled deeper exploration of participant experiences and post course impacts of learning. An exploratory, inductive analysis of the qualitative data was chosen in order to identify themes present and to reduce the impact of preconceived ideas or experiences of the researchers [33]. 

#### 2.4.1. Discussion Forums

All comments made by learners in the course step labelled ‘End of Course Discussion’ were extracted for analysis from (July 2019, October 2019, March 2020 and June 2020). These discussion forums asked learners to share the impacts the course had to their dietary habits and their experiences in taking part in the course (Table 1). While analysis of discussion forum conversations is observational, learners were notified upon registration to Future Learn that comments made in courses may be used for research purposes.

#### 2.4.2. Interviews

To complete in depth qualitative data from a sample of learners, all learners from the March and June 2020 runs of the MOOC were emailed in July 2020, offering an opportunity to take part in a semi-structured interview to explore their experiences and impacts of the MOOC. Learners were emailed a screening questionnaire to ascertain age, discussion forum participation, course satisfaction levels and gender. Purposeful sampling was employed to ensure a range of course satisfaction and discussion forum engagement levels [34]. A flexible and iterative interview guide was developed with questions aligning with levels 1–3 of NWKM (Table 1). Interviews were conducted by MA and CB between September and December 2020 by two dietitian researchers who were familiar with the content and delivery of the MOOC. Interviews were conducted via Zoom (2020 Zoom Inc., San Jose, USA) for 1 h and audio recorded. Specific informed consent was required from each participant prior to the telephone interview. 

Ethics approval by Monash University Human Research Ethics Committee (Ethics number: CF16/905)

### 2.5. Data Analysis

Survey responses were automated and collected using Qualtrics (Qualtrics, Sydney, Australia, 2020). Quantitative analysis of survey data was completed using SPSS Statistics (SPSS version 28, IBM, Chicago, IL, USA, 2021). Descriptive statistics were used to describe participant nutrition information background, with Chi-squared analysis used to explore the relationships between survey questions. Qualitative data were analysed using NVIVO (NVivo version 12, QSR, Melbourne, Australia, 2018) with open, inductive coding conducted. Independent coding of a subset of both discussion forum and interview data was conducted by three researchers and coding was discussed. One researcher then coded all data, with a second researcher independently reviewing final codes. One researcher then developed categories and themes, with independent review, discussion and theme development with a second researcher (Appendix D Table A5). Themes were then aligned to NWKM. Categories and themes which were not relevant to the research question or NWKM were set aside.

### 2.6. Reflexivity 

All researchers were experienced dietitians and researchers, with three out of the four researchers involved in the development and delivery of the MOOC, including appearing in course videos. Researchers discussed how to approach the interviews to help ensure participants felt comfortable in discussing their dietary behaviours with the course presenters. Dietitian researchers refrained from providing individualised dietetic advice during interviews. An inductive approach to qualitative data analysis was taken to try and minimize any assumptions or anecdotal observations from the researchers influencing the outcomes of the research, especially in regard to the positive impacts the MOOC may have had on participants. The research team consciously attended to positive, neutral and negative data points with continuous discussion and scrutiny.

## 3. Results

This section may be divided by subheadings. It should provide a concise and precise description of the experimental results, their interpretation, as well as the experimental conclusions that can be drawn.

### 3.1. Participants

Four thousand, nine hundred and forty-one participants (*n* = 4941) provided consent for their responses to be used in this analysis, with 1003 post course surveys completed. The age of survey participants was comparable to the average age of the 2020 course participants, with an even representation across all age groups from 18+years (Appendix B Table A1). The MOOC reached a global audience with participants from over 150+ countries in each run of the course (Appendix B Table A2). Twenty-four percent of the survey participants indicated their self-perceived baseline level of nutrition knowledge was either non-existent or minimal, 39% indicated they had a basic level of nutrition knowledge, while 32% indicated they have a good general knowledge of nutrition. (Table 2). While 76% of survey respondents indicated they knew at least the basics of nutrition knowledge (with 37% of these indicating they had a good or very good knowledge), 62% of respondents indicated they had not previously studied nutrition prior to enrolling in the MOOC (Table 3). Aligned with the concept of free and open access to knowledge, the Food as Medicine MOOC was marketed to the general public without any prerequisite knowledge on nutrition. However, 10.4% of respondents indicated they had studied nutrition at university or a research institute.

There were 647 comments, made by 585 learners, extracted from the final discussion forum across the four runs of the MOOC. Two hundred and sixty-two learners completed the screening questionnaire offering their participation in the individual semi-structured interviews. Fifty-two learners (with a range of satisfaction levels, genders, and discussion forum participation) were invited to take part. Of these, 12 people were interviewed after completing consent forms (Appendix B Table A3). 

### 3.2. Mapping to NWKM

Quantitative data gathered from surveys were mapped to the relevant levels of NWKM. Thematic analysis of qualitative data identified eight themes regarding learner’s experiences of taking part in the MOOC, and any impacts on dietary behaviours and nutrition knowledge. Each theme was mapped to the relevant NWKM level. The themes were: (i) MOOC style learning and content was liked by participants; (ii) MOOC was a ‘trusted source’ of nutrition information; (iii) Learning new information; (iv) Learning from personal experiences and opinions of peers; (v) The impact of participant self-perceived expertise; (vi) Learning autonomously; (vii) Changing dietary behaviours; (viii) Sharing and influencing the behaviours of others. See Table 4 for a summary of the results.

### 3.3. Level 1—Reaction

Participants’ reactions to the course were very positive with the majority (over 95%) satisfied and indicating it was relevant to their goals (Table 5). Participants indicated they found the discussion forums useful, with 82% finding them somewhat or very useful (Table 5). Only 5% indicated they did not read any comments in the discussion forums. There was a small but significant correlation (Rho 0.26, *p* < 0.01) between course satisfaction and reporting the forum discussions useful.

#### 3.3.1. Qualitative Theme (i): MOOC Style Learning and Content Was Liked by Participants

Data generated from the discussion forums and the individual interviews confirmed that participants were satisfied with, appreciated or enjoyed different course attributes. Of note was the positive high quality of videos, presenters, articles, resources and the overall design of the course. They described how they enjoyed the course and which aspects they found most useful.

*“The most important awareness I got was about mindful eating, so that we don’t overeat things.”* Interview participant 11

There were also learners who indicated they will recommend the course to others. Some learners in the discussion forums thanked the course educators and the university for developing and running the course, with some thanking specific mentors and educators by name. *“I will highly recommend this course to my friends and family.”* Discussion forum comment (Learner 00add5e4) 

*“This has been a very interesting, well structured and resourced course, with an impressive range of papers, guidelines and websites to support each subject covered. It’s always better when a FL course has mentor support throughout, and XXXX has been very thorough and diligent in her weekly summaries despite the current difficulties.”* Discussion forum comment (Learner 0845b302)

Learners from both the forums and interviews mentioned how they enjoyed learning from each other and sharing in the discussion forums by reading comments written by other learners. Some learners thanked their peers for their participation in the discussion forums, and for their advice and information. 

*“The best thing of all, though, has been the lively discussions, and I want to thank my fellow participants for raising excellent questions, and sharing so much.”* Interview participant 11

There were a small number of comments, mainly in the discussion forums, suggesting some participants were unsatisfied with the course and described that the course did not meet expectations, the title the MOOC did not reflect its content or that course was aimed at higher socioeconomic populations. 

*“Before starting the course I had ideas of what you might cover during the three weeks and I was expecting information on incorporating herbs and spices into meals and their individual benefits included on the course as they have so many health giving properties.”* Discussion forum comment (Learner a8960701)

#### 3.3.2. Qualitative Theme (ii): MOOC a Trusted Source of Nutrition Information

The MOOC was described as an evidence-based source of information and access to learning from university lecturers was appreciated. University lecturers were viewed as experts, with several interview participants indicating they preferred to learn information from a university which was viewed as trustworthy. Some interview participants also described increased trust due to the references and additional reading, allowing participants to validate the information themselves. 

*“I can look and I can go, “Oh, that’s where they got it from. Okay, that’s fine.” And it made me feel a lot better about what I was reading so then I could continue from that point and trust what I was getting was right. I didn’t have to go over and try to figure out is it true.”* Interview participant 15

### 3.4. Level 2—Learning

Participants described learning new information in the MOOC, from both course educators and peers. When describing their learning, it was common for participants to also describe their commitment, ability and confidence in applying their newly acquired knowledge. The intention to continue studying/learning about the topic of nutrition, food and health was also mentioned by some participants. Ninety-eight percent of survey respondents indicated they had confidence in applying the learnings from the course (Appendix C Table A4).

#### 3.4.1. Qualitative Theme (iii): Learning New Information 

Participants in both interviews and discussion forums described the MOOC as an informative course in which they learnt new information about nutrition, food and health. It was also common in discussion forums for learners to recap and post information they found interesting and new information they learnt in the course. Some participants also described the course provided them with greater awareness or understanding of nutrition information, or it helped to fill gaps in knowledge that they had. The MOOC provided opportunities for both personal and professional learning, with some participants learning information to apply in both their personal lives and their professional roles. *“So that also I got to know what are the vegetarian protein sources. So that also I got a good salad from this MOOC session, what all can be added with vegetables, combined with vegetables, to increase protein intake in vegetarian people, and also the use of yogurt. How we can include yogurt in the recipes, so that their protein intake will increase.”* Interview participant 11

#### 3.4.2. Qualitative Theme (iv): Learning from Peers’ Personal Experiences and Opinions

Participants interacted with each other within discussion forums answering questions, providing advice or opinions, or thanking them for information. Participants in both discussion forums and interviews mentioned their peers provided them with new ideas, perspectives, and information which assisted them in learning. Participants indicated they were drawn to learn information from peers who they could relate too e.g., a similar medical condition. While some participants acknowledged they do not know the background or expertise of peers, learning from peers was still accepted, and in some cases sought after. Peer learning helped translate information from experts by providing practical suggestions. Some participants shared their personal experiences of using food therapeutically in the past, while others reflected on food and the role it plays/played in their personal health conditions. *“So, I found that quite valuable because I wouldn’t really go to support groups or anything like this because in the past that hasn’t really helped my own health. But an education setting, seeing other people’s opinions and relatable, I find that quite, um, yeah valuable. And also people see things slightly in a different way. Makes you think about other things.”* Interview participant 9

*“I liked using the comments and learning from other students, and I heard about some medicinal plant foods in Africa which was new to me.”* Discussion forum comment (Learner 81337860)

#### 3.4.3. Qualitative Theme (v): The Impact of Participant Self-Perceived Expertise

Some participants included statements regarding their level of knowledge and expertise in nutrition which influenced their learning. When describing their expertise, some learners prefaced their comments with a declaration that they already ate a healthy diet, and then proceeded to outline their own dietary patterns they perceived to be healthy. When describing self-perceived knowledge or behaviours, many participants claimed they learnt nothing new from the MOOC. Others admitted that although they already ate healthily, they were still able to learn new information and that the MOOC validated their current knowledge or dietary patterns. Discussion forums tended to generate more absolute statements about self-perceived knowledge than interview data.

*“Thank you for this course. I have to be honest and say I already knew most of the information and it hasn’t really changed the way I eat.”* Discussion forum comment (Learner f0e4d141)

*“I consider myself to have a healthy diet but I can now see I can make a few changes to it by incorporating more seeds and grains. I have tended to avoid these in the past as I was never too sure what to with them. I am now adding golden linseed and chia seed to my daily porridge alongside the blueberries which I always had prior to the course.”* Discussion forum comment (Learner 25f4e4cc)

#### 3.4.4. Qualitative Theme (vi): Learning Autonomously 

Participants demonstrated autonomy over their learning in a variety of ways including the information they decided to engage with, choosing to engage with particular peers and experts, the time they accessed the course and which media they preferred. Participants decided which information they believed was trustworthy and sought different sources of information if they didn’t agree with information presented, or if there was conflict of information between experts and their fellow learners. Participants described conducting their own research and searched their own information sources to determine facts and gain knowledge. Interview participants described that they consciously decided when to participate in discussion forums, when to stay anonymous and appreciated the flexible learning options so they could participate in learning when it suited them. Time constraints affected learning experiences with participants mentioning that although they liked discussion forums, they were time consuming so often did not join in. 

*“You have a choice whether you’re going to leave a comment. You have a choice whether going to read the comments as well. It’s up to you what level of participation you have.”* Interview no.10

### 3.5. Level 3—Behaviour

Participants reported dietary behaviour change as a result of taking part in this MOOC, with discussion forum comments and interviews describing changes they had made, or were intending to make, to their diets. The majority of survey respondents indicated they are likely to apply what they have learnt in the course, with 72% extremely likely and 24% somewhat likely. No survey respondents indicated they were unlikely to apply their learnings. Additionally, 49% of survey participants indicated they were extremely confident in applying learnings, and 48% were somewhat confident (Appendix C Table A4). 

#### 3.5.1. Fruit and Vegetable Survey 

Sixty-two percent of participants indicated they increased their intake of vegetables by at least 1 serve per day, while 41% indicated eating at least one more serve of fruit (Table 6).

#### 3.5.2. Qualitative Theme (vii): Learners Changed Dietary Behaviours 

The MOOC course was described by many participants in both discussion forums and interviews as a motivator or inspiration for making dietary changes. Participants also described participation in the MOOC as the reason they are intending to, trying or have already made changes to their dietary patterns. Participants also described reflecting or becoming aware of personal changes they could make to their own diets and eating patterns to improve their health. 

*“We try to have [beans and legumes] twice a week. And we mainly do, but now I’ve added more for myself personally. What else? Linseed, which was recommended in your course I’ve also added and hopefully that’s helping.”* Interview participant 1 

*“I have changed my eating patterns since starting the course by reducing my sugar intake and drinking more water. I hope to continue this with time and become more healthy with increased exercise.”* Discussion forum comment (Learner aff89aa5)

*“Will try to ensure that I prepare 5 cups of veg/day for hubby and myself.”* Discussion forum comment (Learner 59039e44)

#### 3.5.3. Qualitative Theme (viii): Sharing and Influencing the Behaviours of Others 

MOOC participation influenced dietary behaviours beyond the participants themselves, with some participants describing sharing and discussing the information they learnt with their family, friends and colleagues. Others described how they now influence their family dietary behaviours and how they have changed since they participated in the course. 

*“I loved finding out why we need to eat certain foods and relaying this information to my kids in a slightly more kid friendly version! I have found they are more engaged at meal times when they understand why and what they are eating.”* Discussion forum comment (Learner 5321236b)

### 3.6. Level 4—Results 

The overarching principal and aim of delivering the Food as Medicine MOOC were to nudge participant’s dietary behaviours and patterns to be aligned with the Australian Dietary Guidelines [35]. In addition, while this evaluation research was not set up to specifically measure Level 4 of NWKM, the results from the self-reported fruit and vegetable survey provided some evidence that demonstrates the MOOC’s overarching purpose may have been met. Self-reported daily vegetable intake indicated 38% of participants were meeting the current Australian Dietary Guideline (ADG) recommendations for vegetable intake after taking part in the MOOC. This is above the current percentage of the Australian population meeting the recommended 5 serves (Table 7). Sixty-six percent of participants self-reported fruit intake met the recommendations which is above the Australian average of 50% (Table 7).

## 4. Discussion

To our knowledge, this is the first comprehensive mixed methods evaluation of a nutrition focused MOOC, using published MOOC evaluation protocols and aligning with the NWKM for learning and impact evaluation [21,22]. The evaluation used Levels 1–3 of the NWKM and demonstrated the MOOC met its aim for participants involved in this research. They learnt new information and reported behaviour change consistent with principles outlined in global government dietary guidelines [35,37,38]. Triangulating multiple data sources, including RWD from MOOC discussion forums, enabled a comprehensive picture of participant experiences. Each participant had an individual experience. Quantitative data was positive overall, with participants’ qualitative responses from discussion forums and interviews mostly consistent with this. Participants described their own learning and interpretations of the course content and often supported each other throughout the course.

The MOOC reached a global and diverse audience with participants from a wide range of adult ages and baseline levels of nutrition knowledge. This demonstrated that MOOCs may be potentially powerful communication tools for nutrition and food professionals to harness due to their ability to reach large, global audiences [17,19,39]. This reach and impact may be broader than more traditional methods of communicating and educating on nutrition such as group sessions or presentations, print and TV media, or public health campaigns. Our research demonstrated at least 673 people increased their serves of vegetables by at least one serve per day after participation in the course, which has been shown to reduce risk of chronic disease and improve health [40]. Increasing intake of vegetables and fruits by at least 1 serve per day has been outlined in a range of government dietary guidelines to have protective effects for a range of chronic health diseases such as heart disease, stroke, some cancers and type 2 diabetes [40,41]. 

Providing nutrition information in an open forum like a MOOC presents challenges for nutrition professionals due to learners having differing levels of knowledge, experiences and learning preferences. Participants with different baseline levels of knowledge and goals for learning may impact the way in which they react to information being provided [42]. Despite the diverse audience, our findings show the reaction of participants was positive, with the majority being satisfied with the MOOC, and describing the course as relevant to their own personal goals. This may be because learners were not just learning information from educators, but also from each other and benefiting from sharing experiences with their peers. 

MOOC’s flexible learning environment and large cohorts align with the public’s greater need for learning autonomy and information to which they can relate [43,44]. This is consistent with our findings which highlight that autonomy over learning was important. With the public taking greater charge of their learning, robust evaluation of publicly available nutrition information sources, such as MOOCs, is crucial for nutrition professionals to understand whether they are providing effective education and meeting learners expectations. The discussion forum data utilised in conjunction with the parameters set in recent published MOOC protocols, allowed for a greater exploration of learners’ experiences of the course and their learning preferences [21,22]. This research suggests that the expert didactic lectures are enhanced when supplemented with online interaction between peers [45]. 

Respect for learners, their prior knowledge and learning preferences is paramount and consistent with adult learning theory principles [43,44]. Our findings highlight 61% of participants indicated they had no previous nutrition training, but 76% reported they had at least basic nutrition knowledge or higher. Discussion forums also revealed participants’ self-perceived nutrition knowledge to be high but admitting to new learning or that the MOOC provided confirmation of their current practice. Online learning environments, such as MOOCs, that include both passive and active learning features can assist in providing for many different adult learning preferences [46]. This may assist nutrition professionals reach and resonate with people who have high self-perceived levels of nutrition knowledge who may be harder to influence [47]. 

Previous research has demonstrated that people consume nutrition information authored by people with a range of education backgrounds, including people with no nutrition qualifications [1,11,12]. Similarly, our research showed learners felt comfortable consuming information from peers, regardless of background, with 82% of learners indicating the discussion forums were a useful part of the course. There was also a relationship between course satisfaction and usefulness of discussion forums, suggesting peer interaction was an important part of the learning experience. Learning from different cultural food experiences of peers was valued, suggesting a global cohort with diverse backgrounds, may be beneficial for learning [7,24]. Practical and often personalised suggestions made by peers in discussion forums helped participants apply recommendations made by educators in the course. This highlights the need for experts to be open-minded to interactions and discussions between peers, sometimes filling the role of facilitator, with the role of teacher being briefly passed on to peers to assist with the implementation of expert advice. 

MOOC participation appeared to influence dietary behaviours, with many people indicating they changed their dietary behaviours, were trying to make changes, or were intending to make changes. These descriptions are consistent with the Transtheoretical Model of behaviour change, which recognises people are at different stages of readiness to change [48,49]. Because of the diversity in participants’ backgrounds and circumstances, it is likely participants will be at different points on their diet, nutrition and health journeys, and hence different stage of behaviour change. The flexible pedagogical structure of MOOCs provides experts with a platform for educating large cohorts of learners while also affording participants with different backgrounds and learning needs the ability to choose information relevant to their learning [50].

### Strengths and Limitations

Strengths of this analysis include the range of quantitative and qualitative evaluation measures used. Although we do not have data from all learners, Real World Data, such as the data from discussion forums, is valuable as it can capture a large number of people’s voices [51]. The ability to analyse user-generated data from discussion forums allowed for qualitative interrogation of a larger sample size of learners than interview methods alone would have allowed. The global sample of learners, along with their diverse demographics was also a strength to help act as a proxy for the general population. Limitations related to real world research for this large global sample included individual participants could not be tracked throughout the course, nor was there a control group. The course was ‘real world’, with learners participating in the MOOC freely, not specifically to be part of research. Therefore, mandatory participation under controlled research conditions, was not possible i.e., detailed, matched pre and post course surveys. Not all MOOC learners completed the surveys or commented in discussion forums, thus only a sample of participants’ experiences was explored; however, the addition of discussion forum data helped alleviate this potential bias. As participation in discussion forums, surveys and interviews was not mandatory, there was a risk that either only highly motivated or disgruntled participants engaged with the research. Apart from pre-course survey data, the other data were collected from participants reaching the end of the MOOC. This may mean they were more likely to have been satisfied with the course because they continued to take part until the end. MOOC completion rates are notoriously low, so evaluation measures early in the course would assist in capturing the experiences of participants who may be less satisfied [52]. Although we do not have data from all learners, this research brings deeper understanding to the impact on learners of a nutrition focussed MOOC. Social desirability bias may have impacted the responses to semi-structured interviews as one researcher was involved in delivery of the MOOC [53]. Further evaluation with the REAIM framework is suggested to build upon the Kirkpatrick evaluation to explore delivery, sustainability, and maintenance of the MOOC to determine whether MOOCs are a viable way to deliver health promotion information and disseminate knowledge more broadly [54]. 

## 5. Conclusions

This comprehensive evaluation has explored participants’ experiences of a nutrition focused MOOC, highlighting they were generally positive, with participants learning and sharing information, and with some reporting a change to their dietary behaviour. MOOCs have the potential for nutrition professionals to effectively communicate evidence-based nutrition information to large, global audiences. They can provide a middle ground, where nutrition professionals share evidence-based information and facilitate discussions, while the adult learners can play a role and maintain their learning autonomy and preferences. 

## Figures and Tables

**Figure 1 nutrients-14-03680-f001:**
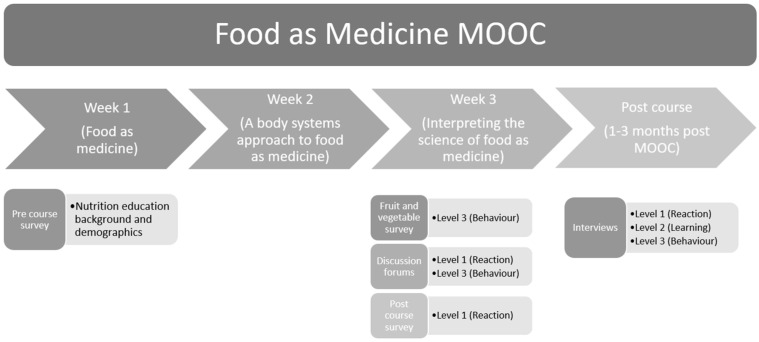
Data collection sources from different stages of the MOOC.

**Table 1 nutrients-14-03680-t001:** Evaluation methods and corresponding levels of NWKM.

NWKM Levels	Level Description	Questions Asked	Method of Evaluation
**Level 1: Reaction**	Satisfaction	How satisfied were you with the course?	Quantitative post course survey imbedded in final step of the MOOC
	Engagement	Did you find the discussion forums in the course useful?	
	Relevance	How relevant was the course content for you and your goals?	
	Satisfaction	What did you like most about the course? What did you think about the presence of discussion forums in the MOOC?	Comments in final discussion forum extracted for qualitative thematic analysis;
	Engagement	Consider sharing with other learners some of the ways your food intake or approach to eating has changed, and how you’re now using food as medicine. You might like to take some time to read comments made by other learners, and if you find these comments interesting, respond to them.	
	Satisfaction	What did you think of the MOOC? What did you think about the presence of discussion forums in the MOOC?	Semi-structured interviews conducted with MOOC participants, with qualitative thematic analysis conducted
	Engagement	How active were you in the discussion forums?	
	Relevance	Why did you/did you not use the discussion forums?How did you react when you read something that they didn’t agree with in discuss forums? How do you view the information provided by other learners in discussion forums? How likely are you follow or believe what they say?	
**Level 2: Learning**	Commitment	How likely are you to apply what you have learnt?	Quantitative post course survey imbedded in final step (on completion) of the MOOC
	Confidence	How confident are you in applying what you have learnt?	
	Commitment	How likely are you to continue seeking further information on nutrition?	
	What affect has the MOOC had on your beliefs/knowledge around nutrition? What new information have you learnt from the MOOC? Can you describe any changes to your views on nutrition and food topics the MOOC has had? Where the discussion forums useful for your learning?	Knowledge	Semi-structured interviews conducted with purposefully sampled participants, with qualitative thematic analysis conducted
	Attitude	How important are credentials to you when learning information? How do you view the information provided by other learners in discussion forums? Do you believe it? Do you tend to follow what they say?	
**Level 3:** **Behaviour**	Behaviours: application or utilisation of what was learnt?	What has been the biggest change you have made since participating in the MOOC? Describe any impact on other members of your household taking part in the MOOC has had. How has your participation in the MOOC affected your eating habits? Describe any impacts taking part in the MOOC has had on your eating habits?	Semi-structured interviews conducted with MOOC participants, with qualitative thematic analysis
		How have your food choices changed? What are you doing to eat more healthily? How are you now using food as medicine? How has your food intake or approach to eating changed?	Comments in final discussion forum extracted for qualitative thematic analysis
		After completing this course, how many serves of vegetables do you now eat each day? In total, how many serves of vegetables do you usually eat each day? After completing this course, how many serves of fruit do you now eat each day? In total, how many serves of fruit do you usually eat each day?	Quantitative survey measuring self-reported fruit and vegetable intake

**Table 2 nutrients-14-03680-t002:** Self-perceived level of participant nutrition knowledge before commencing the MOOC.

Self-Perceived Level of Nutrition Knowledge	*n* = 4941	Percent
I have no knowledge about nutrition	128	2.6
I have only a little knowledge about nutrition	1042	21.1
I know the basics of nutrition information	1930	39.1
I have a good general knowledge of nutrition	1617	32.7
I am very knowledgeable about nutrition	224	4.5

**Table 3 nutrients-14-03680-t003:** Participant reported highest level of studying nutrition before commencing the MOOC.

Highest Level of Nutrition Study	*n* = 4941	Percent
University or research institute	514	10.4
TAFE of college	226	4.6
High School	341	6.9
Short course	813	16.5
No formal nutrition study	3047	61.7

**Table 4 nutrients-14-03680-t004:** Summary of quantitative and qualitative results mapped to NWKM.

NWKM Level	Results
Level 1: Reaction	959 (95.6%) participants indicated they were extremely or somewhat satisfied975 (97.2%) participants indicated the course was very or somewhat relevant to learning goals
	Qualitative Theme (i): MOOC style learning and content was liked by participants.
	Qualitative theme (ii): MOOC a trusted source of nutrition information.
Level 2: Learning	Qualitative theme (iii): Learning new information
	Qualitative theme (iv): Learning from personal experiences and opinions of peers.
	Qualitative theme (v): The impact of participant self-perceived expertise.
	Qualitative theme (vi): Learning autonomously.
Level 3: Behaviour	96% of participants indicated they were extremely likely or somewhat likely to apply what they have learnt
	97% of participants indicated they were extremely of somewhat confident in applying learning
	62% of participants reported having at least 1 more serve of vegetables each day
	Qualitative theme (vii): Learners changed dietary behaviours.
	Qualitative theme (viii): Sharing and influencing the behaviours of others.

**Table 5 nutrients-14-03680-t005:** Participant reaction to the MOOC.

	Very Positive	Somewhat Positive	Neither	Somewhat Negative	Extremely Negative
How satisfied are you with the course? * (*n* = 1003)	642 (64.0%)	317 (31.6%)	36 (3.6%)	5 (0.5%)	3 (0.3%)
How relevant was the course for you and your goals? ** (*n* = 1003)	600 (59.8%)	375 (37.4%)	25 (2.5%)	3 (0.3%)	0 (0%)
Did you find the discussion forums in the course useful? *** (*n* = 1003)	295 (29.4%)	535 (53.3%)	115 (11.5%)	5 (0.5%)	53 (5.3%)

* measured by satisfaction levels: extremely satisfied, somewhat satisfied, neither satisfied or dissatisfied, somewhat dissatisfied or extremely dissatisfied. ** measured by relevance levels: very relevant, somewhat relevant, not relevant or irrelevant, somewhat irrelevant, very irrelevant. *** measured by usefulness: very useful, somewhat useful, not very useful, definitely not useful, I did not read any comments in the discussion forums.

**Table 6 nutrients-14-03680-t006:** Participant-reported intake of fruit and vegetables since doing the MOOC.

	Serves of Vegetables and Fruit	Vegetable Intake *n* = 1090	Fruit Intake *n* = 1088
**Change of serves**	Eating 1–2 serves more	463 (42.5%)	326 (30.0%)
	Eating 2+ serves more	210 (19.3%)	118 (10.8%)
	Eating the same	388 (35.6%)	576 (52.9%)
	Eating 1–2 serves less	18 (1.7%)	52 (4.8%)
	Eating 2+ serves less	11 (1.0%)	16 (1.5%)
**Total number of serves per day**	0–1	39 (3.6%)	27 (2.5%)
	2	122 (11.2%)	168 (15.4%)
	3	207 (19.0%)	715 (65.6%)
	4	309 (28.3%)	178 (16.3%)
	5	243 (22.3%)	N/A
	6 or more	168 (15.4%)	N/A

**Table 7 nutrients-14-03680-t007:** Participants meeting recommended fruit and vegetables intake since doing the MOOC.

	Vegetable Intake *n* = 1088	Fruit Intake *n* = 1088
Meeting Australian Dietary Guidelines recommendation after participating in course	37.8%	65.7%
Current proportion of Australian’s meeting the recommended intake [36]	<7%	50%

## Data Availability

Not applicable.

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
