# Peer review of "Exploring Impacts of a Nutrition-Focused Massive Open Online Course"

_nutrients, 2022, doi:10.3390/nu14183680_

Round 1

Reviewer 1 Report

The paper describes the process of evaluation of Massive Open Online Courses (MOOCs) in the field of nutrition.

MOOCs enable general public access to knowledge provided by academic experts, however with possibility of peer-to-peer sharing of information and learning from each other as opposite to traditional didactic mode of education.

To evaluate the set of nutritional courses, the Kirkpatrick Model was selected which includes four levels of assessment (Reaction, Learning, Behaviour and Results). The three first levels were used by the authors. Data from discussion forums,  the results of semi-structured interviews and purpose built surveys were taken into account also.

The evaluated courses took place in 2020. About 45 th. participants signed up to the MOOC and about 33 th. were learning. However only about 5 th. answered pre-course survey and about 1 th. – post-course survey. Twelve participants participated in the semi-structured interviews.

The authors present descriptive results obtained from the surveys as well as some content of the questionnaires. Selected opinions from the qualitative themes are cited.

The study results are distorted by strong selection bias. Based on these so selective results, the conclusions are not valid, in my opinion.

Author Response

Reviewer 1

Thank you for taking the time to review our manuscript – we appreciate your feedback and comments. Please see below for responses to your reviewer comments.

  1. The paper describes the process of evaluation of Massive Open Online Courses (MOOCs) in the field of nutrition.
  2. MOOCs enable general public access to knowledge provided by academic experts, however with possibility of peer-to-peer sharing of information and learning from each other as opposite to traditional didactic mode of education.
  3. To evaluate the set of nutritional courses, the Kirkpatrick Model was selected which includes four levels of assessment (Reaction, Learning, Behaviour and Results). The three first levels were used by the authors. Data from discussion forums,  the results of semi-structured interviews and purpose built surveys were taken into account also.
  4. The evaluated courses took place in 2020. About 45 th. participants signed up to the MOOC and about 33 th. were learning. However only about 5 th. answered pre-course survey and about 1 th. – post-course survey. Twelve participants participated in the semi-structured interviews.
  5. The authors present descriptive results obtained from the surveys as well as some content of the questionnaires. Selected opinions from the qualitative themes are cited.
    • The selected quotes were chosen to offer readers authentic examples of participants’ discussions, were illustrative of the relevant theme and represented patterns found in the data. (Lingard, L. Beyond the default colon: Effective use of quotes in qualitative research. Perspect Med Educ 8, 360–364 (2019)).
  6. The study results are distorted by strong selection bias. Based on these so selective results, the conclusions are not valid, in my opinion.
    • Thank you for your feedback – we welcome your input so we can reflect on our research and manuscript. We agree there is bias in the results due to the sample of participants who took part in the research and discussion forums, and we highlighted a number of these in the limitations section including the challenges of using Real World Data (RWD) and participation rates of MOOCs. Reflecting on your feedback, we agree this could be clarified further and so have added additional discussion around this in the limitations (see lines 597-603). We have also amended the introduction of the discussion to more clearly articulate that conclusions are based on participants who were motivated to reach the end of the course (See lines 503-504), as well as rephrased the conclusion (lines 612-614). Although we do not have data for all learners, Real World Data, such as the data found in discussion forums, is valuable as it can capture a large number of people’s voices, despite challenges regarding representativeness (McDonald, L., Malcolm, B., Ramagopalan, S. et al. Real-world data and the patient perspective: the PROmise of social media?. BMC Med 17, 11 (2019)). We agree using Real World Data means we are only going to get data that was naturally provided by participants, which is why we triangulated the data with survey and interview data. The sampling method we employed for interview participants endeavoured to address this, in that we selected participants who reported a range of levels of satisfaction (extremely dissatisfied, somewhat dissatisfied, somewhat satisfied and extremely satisfied)

Reviewer 2 Report

 "Exploring impacts of a nutrition-focused Massive Open Online Course". This is a well-researched, prepared, and presented manuscript evaluating an innovative approach to nutrition education for the public. 

Abstract: Sound (minor point) Line 21: omit and ie  semi-structured interviews and were thematically analysed

Introduction: Sound 

Materials and Methods: Sound (minor point) Line 249: suggest their self- perceived baseline level of nutrition knowledge. In Results, Line 402, you use "When describing self-perceived knowledge or behaviours,". Be consistent.

Results: Sound. A minor point related to the Materials and Methods point, should Table 2 be labelled; "Participants' self-perceived level of nutrition knowledge before commencing the MOOC".? Also, should the first subheading be labelled, 'Self-perceived level of nutrition knowledge? 

Should the headings in Table 1 Reaction be flipped to match the Satisfaction, Engagement, and Relevance headings below?

Discussion: Sound

References: Sound (minor point) Reference 3 has a full title whereas, other references have abbreviated titles. Please amend.

Author Response

Thank you for taking the time to review our manuscript – we appreciate your feedback and comments. Please see below for responses to your reviewer comments.

  1. "Exploring impacts of a nutrition-focused Massive Open Online Course". This is a well-researched, prepared, and presented manuscript evaluating an innovative approach to nutrition education for the public. 
    • Thank you for the feedback
  2. Abstract: Sound (minor point) Line 21: omit and ie semi-structured interviews and were thematically analysed
    • Agreed - changed
  3. Introduction: Sound 
  4. Materials and Methods: Sound (minor point) Line 249: suggesttheir self- perceived baseline level of nutrition knowledge. In Results, Line 402, you use "When describing self-perceived knowledge or behaviours,". Be consistent.
    • Agreed - changed
  5. Results: Sound. A minor point related to the Materials and Methods point, should Table 2 be labelled; "Participants' self-perceived level of nutrition knowledge before commencing the MOOC".? Also, should the first subheading be labelled, 'Self-perceived level of nutrition knowledge? 
    • Agreed - changed
  6. Should the headings in Table 1 Reaction be flipped to match the Satisfaction, Engagement, and Relevance headings below?
    • Agreed - we have made the headings in column two consistent in terms of order with the other rows in Reaction section i.e. in the order of Satisfaction, Engagement and Relevance.
  7. Discussion: Sound
    • Thank you
  8. References: Sound (minor point) Reference 3 has a full title whereas, other references have abbreviated titles. Please amend.
    • Have changed the journal title to Int J Med Educ

Round 2

Reviewer 1 Report

Unfortunately, I'm not convinced by the authors' response. It does not change the fact that the study has huge selection bias. Nothing has been done at the design phase  of the study to reduce or even address this potential, well known problem of using surveys to assess e-learning.